# SHAPLEY EXPLAINABILITY ON THE DATA MANIFOLD

**Christopher Frye,**    **Damien de Mijolla,**    **Tom Begley,**
**Laurence Cowton,**    **Megan Stanley,**    **Ilya Feige**

Faculty, 54 Welbeck Street, London, UK

## ABSTRACT

Explainability in AI is crucial for model development, compliance with regulation, and providing operational nuance to predictions. The Shapley framework for explainability attributes a model's predictions to its input features in a mathematically principled and model-agnostic way. However, general implementations of Shapley explainability make an untenable assumption: that the model's features are uncorrelated. In this work, we demonstrate unambiguous drawbacks of this assumption and develop two solutions to Shapley explainability that respect the data manifold. One solution, based on generative modelling, provides flexible access to data imputations; the other directly learns the Shapley value-function, providing performance and stability at the cost of flexibility. While "off-manifold" Shapley values can (i) give rise to incorrect explanations, (ii) hide implicit model dependence on sensitive attributes, and (iii) lead to unintelligible explanations in higher-dimensional data, on-manifold explainability overcomes these problems.

## 1 INTRODUCTION

Explainability in AI is central to the practical impact of AI on society, thus making it critical to get right. While many dichotomies exist within the field — between *local* and *global* explanations (Ribeiro et al., 2016), between *post hoc* and *intrinsic* interpretability (Rudin, 2019), and between *model-agnostic* and *model-specific* methods (Shrikumar et al., 2017) — in this work we focus on local, post-hoc, model-agnostic explainability as it provides insight into individual model predictions, does not limit model expressiveness, and is comparable across model types.

In this context, explainability can be treated as a problem of attribution. Shapley values (Shapley, 1953) provide the unique attribution method satisfying a set of intuitive axioms, e.g. they capture all interactions between features and sum to the model prediction. The Shapley approach to explainability has matured over the last two decades (Lipovetsky & Conklin, 2001; Kononenko et al., 2010; Štrumbelj & Kononenko, 2014; Datta et al., 2016; Lundberg & Lee, 2017).

Implementations of Shapley explainability suffer from a problem common across model-agnostic methods: they involve marginalisation over features, achieved by splicing data points together and evaluating the model on highly unrealistic inputs (e.g. Fig. 1). Such splicing would only be justified if all features were independent; otherwise, spliced data lies *off the data manifold*.

Outside the Shapley paradigm, emerging explainability methods have begun to address this problem. See e.g. Anders et al. (2020) for a general treatment of the off-manifold problem in gradient-based explainability. See also Chang et al. (2019) and Agarwal et al. (2019) for image-specific explanations that respect the data distribution.

Within Shapley explainability, initial work towards remedying the off-manifold problem has emerged; e.g. Aas et al. (2019) and Sundararajan & Najmi (2019) explore empirical and kernel-based estimation techniques, but these methods do not scale to complex data. A satisfactorily general and performant solution to computing Shapley values *on the data manifold* has yet to appear and is a focus of this work. Our main contributions are twofold:

- Sec. 3 compares on- and off-manifold explainability, focusing on novel and unambiguous shortcomings of off-manifold Shapley values. In particular, we show that off-manifold explanations are often incorrect, and that they can hide implicit model dependence on sensitive features.

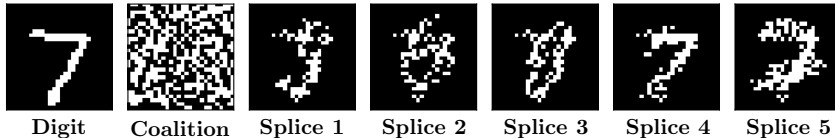

| Digit | Coalition | Splice 1 | Splice 2 | Splice 3 | Splice 4 | Splice 5 |

Figure 1: An MNIST digit, a coalition of pixels in a Shapley calculation, and 5 off-manifold splices.

- Sec. 4 develops two methods to compute on-manifold Shapley values on general data sets: (i) a flexible generative-modelling technique to learn the data's conditional distributions, and (ii) a simple supervised-learning technique that targets the Shapley value-function directly. We demonstrate the effectiveness of these methods on higher-dimensional data with experiments.

## 2 BACKGROUND ON SHAPLEY EXPLAINABILITY

The *Shapley value* (Shapley, 1953) is a method from cooperative game theory that distributes credit for the total value $v(N)$ earned by a team $N = \{1, 2, \ldots, n\}$ among its players:

$$\phi_v(i) = \sum_{S \subseteq N \setminus \{i\}} \frac{|S|! \, (n - |S| - 1)!}{n!} \left[ v(S \cup \{i\}) - v(S) \right] \tag{1}$$

where the value function $v(S)$ indicates the value that a coalition of players $S$ would earn without their other teammates. The Shapley value $\phi_v(i)$ represents player $i$'s marginal value-added upon joining the team, averaged over all orderings in which the team can be constructed.

In supervised learning, let $f_y(x)$ be a model's predicted probability that data point $x$ belongs to class $y$.[1] To apply Shapley attribution to model explainability, one interprets the features $\{x_1, \ldots, x_n\}$ as players in a game and the output $f_y(x)$ as their earned value. To compute Shapley values, one must define a value function representing the model's output on a coalition $x_S \subseteq \{x_1, \ldots, x_n\}$.

As the model is undefined on partial input $x_S$, the standard implementation (Lundberg & Lee, 2017) samples out-of-coalition features, $x'_{\bar{S}}$ where $\bar{S} = N \setminus S$, unconditionally from the data distribution:

$$v^{(\text{off})}_{f_y(x)}(S) = \mathbb{E}_{p(x')} \left[ f_y(x_S \sqcup x'_{\bar{S}}) \right] \tag{2}$$

We refer to this value function, and the corresponding Shapley values, as lying *off the data manifold* since splices $x_S \sqcup x'_{\bar{S}}$ generically lie far from the data distribution. Alternatively, conditioning out-of-coalition features $x'_{\bar{S}}$ on in-coalition features $x_S$ would result in an *on-manifold* value function:

$$v^{(\text{on})}_{f_y(x)}(S) = \mathbb{E}_{p(x'|x_S)} \left[ f_y(x') \right] \tag{3}$$

The conditional distribution $p(x'|x_S)$ is not empirically accessible in practical scenarios with high-dimensional data or many-valued (e.g. continuous) features. A performant method to compute on-manifold Shapley values on general data is until-now lacking and a focus of this work.

Shapley values $\phi_{f_y(x)}(i)$ provide *local* explainability for the model's prediction on data point $x$. To understand the model's global behaviour, one aggregates the $\phi_{f_y(x)}(i)$'s into *global Shapley values*:

$$\Phi_f(i) = \mathbb{E}_{p(x,y)} \left[ \phi_{f_y(x)}(i) \right] \tag{4}$$

where $p(x, y)$ is the labelled-data distribution. Global Shapley values can be seen as a special case of the global explanation framework introduced by Covert et al. (2020). As a consequence of the axioms (Shapley, 1953) satisfied by the $\phi_{f_y(x)}(i)$'s, global Shapley values satisfy a sum rule:

$$\sum_{i \in N} \Phi_f(i) = \mathbb{E}_{p(x,y)} \left[ f_y(x) \right] - \mathbb{E}_{p(x')} \mathbb{E}_{p(y)} \left[ f_y(x') \right] \tag{5}$$

One interprets the global Shapley value $\Phi_f(i)$ as the portion of model accuracy attributable to the $i^{\text{th}}$ feature. Indeed, the first term in Eq. (5) is the accuracy one achieves by sampling labels from $f$'s predicted probability distribution over classes. The offset term, which relates to class balance, is not attributable to any individual feature.

---

[1] The results of this paper can be applied to regression problems by reinterpreting $f_y(x)$ as the model's predicted value rather than its predicted probability.

## 3  EVIDENCE IN FAVOUR OF ON-MANIFOLD EXPLAINABILITY

The key differences between on- and off-manifold Shapley values is a subject of ongoing discussion; see Sundararajan & Najmi (2019) or Chen et al. (2020) for recent overviews. Here we focus on theoretical arguments and experimental evidence yet to appear in the literature, in favour of the on-manifold approach. We begin with mathematically precise differences between on- and off-manifold methods in Sec. 3.1 and present unambiguous drawbacks of off-manifold Shapley values in Sec. 3.2.

### 3.1  ON- VERSUS OFF-MANIFOLD DIFFERENCES MADE PRECISE

Suppose the model's input features $x_1, \ldots, x_n$ are the result of a data-generating process seeded by unobserved latent variables $z_1, \ldots, z_d$. Then there exist functional relationships

$$x_i = g_i(z_1, \ldots, z_d; \epsilon_i) \quad \text{for} \quad i = 1, \ldots, n \tag{6}$$

where $\epsilon_i$ represents noise in $x_i$. In the limit of small $\epsilon_i$'s, there are $d$ directions in which a data point $(x_1, \ldots, x_n)$ can be perturbed while remaining consistent with the data distribution: these correspond to perturbations in $z_1, \ldots, z_d$ in Eq. (6). The data thus lives on a $d$-dimensional manifold in ambient $n$-dimensional features space, and therefore satisfies $n - d$ constraints on the $x_i$'s:

$$\Psi_k(x_1, \ldots, x_n) = 0 \quad \text{for} \quad k = 1, \ldots, n - d \tag{7}$$

On-manifold Shapley values evaluate the model on inputs that satisfy these constraints, while the off-manifold approach uses spliced data that generically break them. For a more detailed and mathematically precise discussion of the data manifold in this context, see Anders et al. (2020).

ALGEBRAIC MODEL DEPENDENCE CAN BE MISLEADING

Any model $f_y(x)$ can be written in many algebraic forms that all evaluate identically on the data manifold. To show this, one can add any of the $n - d$ constraints from Eq. (7) to any of the model's $n$ input slots. This changes the model's algebraic form but does not affect the model's output on the data, since each constraint equals zero on-manifold. The model $f_y(x)$ thus belongs to an $n(n - d)$ dimensional equivalence class of functions that behave indistinguishably on the data.

On-manifold Shapley values provide the same explanation for any two models that evaluate identically on the data distribution, because on-manifold explanations do not involve evaluation anywhere else. Off-manifold Shapley values provide different explanations for two models in the same equivalence class, as spliced data in the off-manifold value function break the constraints of Eq. (7).

HIDDEN MODEL DEPENDENCE ON SENSITIVE ATTRIBUTES

This is not an academic concern: it follows that off-manifold explanations are vulnerable to adversarial model perturbations that hide dependence on select input features (Dombrowski et al., 2019; Slack et al., 2020). Dimanov et al. (2020) demonstrated that the off-manifold Shapley value for a sensitive feature like gender could be reduced near zero via this vulnerability.

To see how this can happen, suppose that input feature $x_1$ represents gender and formally solve one of the constraints in Eq. (7) for $x_1$. The result, say $x_1 = \tilde{\Psi}(x_2, \ldots, x_n)$, can then be used to transform any model $f_y(x)$ into another

$$\tilde{f}_y(x_2, \ldots, x_n) = f_y\big(\tilde{\Psi}(x_2, \ldots, x_n), x_2, \ldots, x_n\big) \tag{8}$$

that has no algebraic dependence on gender $x_1$ but behaves identically to $f_y(x)$ on the data manifold. The off-manifold Shapley value for gender in $\tilde{f}$ would vanish, since the off-manifold value function of Eq. (2) depends on $x_1$ only through $\tilde{f}$ (i.e. not at all). This result is problematic, since the two models behave equivalently on the data and thus possess the same gender bias.

In contrast, the on-manifold Shapley values for $f$ and $\tilde{f}$ would be identical, as $x_1$ dependence enters the on-manifold value function of Eq. (3) through the conditional expectation value. In a sense, on-manifold Shapley values represent the model's dependence on the information content of each feature, rather than the model's algebraic dependence.

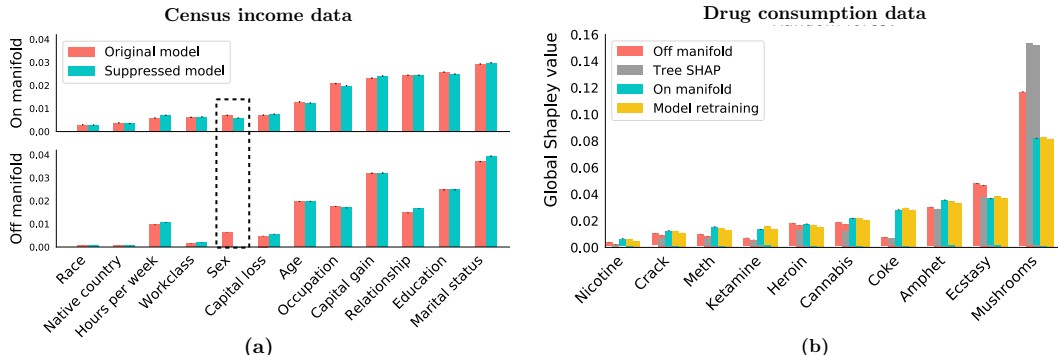

Figure 2: (a) Vulnerability of off-manifold explanations to hidden model dependence. (b) Explanations of a fixed model compared to a model that is retrained on each Shapley coalition of features.

We can demonstrate this on UCI Census Income data (Dua & Graff, 2017). We trained a neural network to predict whether an individual's income exceeds \$50k based on demographic features in the data. Coral bars in Fig. 2(a) display global Shapley values for this "Original model". (On-manifold values were computed with the unsupervised method developed in Sec. 4.1.)

We then trained an alternative model by fine-tuning the neural network above on a loss function that penalises model dependence on sex; see App. B for full details of this experiment. This resulted in a "Suppressed model" that makes identical predictions as the original model on 98.5% of the data. Teal bars in Fig. 2(a) display global Shapley values for this model. Note that the off-manifold Shapley value for sex is zero despite the similar behaviour exhibited by the original and suppressed models on the data. In contrast, on-manifold Shapley values explain both models similarly.

ON-MANIFOLD SHAPLEY VALUES IN THE OPTIMAL-MODEL LIMIT

Here we present a result that strengthens the connection between on-manifold Shapley values and the data distribution: in the limit of an optimal model of the data, on-manifold Shapley values converge to an explanation of how the information in the data associates with the labelled outcomes.

To show why this holds, suppose the predicted probability $f_y(x)$ converges to the true underlying distribution $p(y|x)$. In this optimal-model limit (which is approached in the limit of abundant data and high model capacity) the on-manifold value function of Eq. (3) becomes

$$v^{(\text{on})}_{f_y(x)}(S) \to \int dx'_{\bar{S}} \, p(x'_{\bar{S}}|x_S) \, p(y \mid x_S \sqcup x'_{\bar{S}}) = p(y \mid x_S) \tag{9}$$

which shows that value is attributed to $x_i$ based on $x_i$'s predictivity of the label $y$.

We can demonstrate this on UCI Drug Consumption data (Dua & Graff, 2017). Using the 10 binary features listed in Fig. 2(b) – Mushrooms, Ecstasy, etc. – we trained a random forest $f$ to predict whether individuals had consumed an 11th drug: LSD. As the data contains just 10 binary features, we were able to empirically sample the conditional distributions in the on-manifold value function, Eq. (3). See Fig. 2(b) for the resulting off- and on-manifold global Shapley values.

Next we fit a separate random forest $g_S$ to each coalition $S$ of features, $2^{10}$ models in total, in the spirit e.g. of Štrumbelj et al. (2009). We used the accuracy $A(g_S)$ of each model, in the sense of Eq. (5), as the value function for an additional Shapley computation:

$$\Phi_g(i) = \sum_{S \subseteq N \setminus i} \frac{|S|! \, (n - |S| - 1)!}{n!} \, [A(g_{S \cup i}) - A(g_S)] \tag{10}$$

where $\Phi_g(i)$ is directly the average gain in accuracy that results from adding feature $i$ to the set of inputs. These values are labelled "Model retraining" in Fig. 2(b). Note their agreement with the on-manifold explanation of the fixed random forest $f$. On-manifold Shapley values thus indicate which features in the data are most predictive of the label.

This consistency check allows us to show in passing that Tree SHAP (Lundberg et al., 2018; 2020) does *not* provide a method for on-manifold explainability. Observe in Fig. 2(b) that Tree SHAP

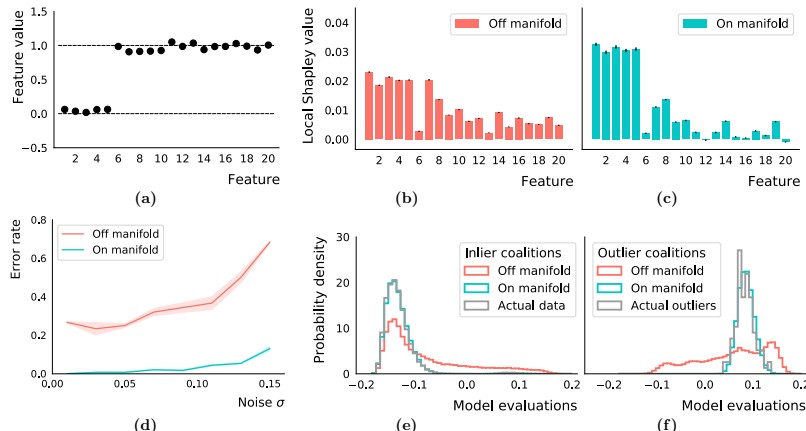

Figure 3: An individual outlier (a), its off- and on-manifold explanations (b & c), the error rate in explanations (d), and the distribution of model outputs on Shapley coalitions (e & f).

roughly tracks the off-manifold explanation, albeit larger on the most predictive feature and somewhat smaller on the others. This occurs because trees tend to split on high-predictivity features first, and Tree SHAP privileges early-splitting features in an otherwise off-manifold calculation.

## 3.2 UNAMBIGUOUS SHORTCOMINGS OF OFF-MANIFOLD EXPLAINABILITY

Whereas above we clarified precise differences between on- and off-manifold Shapley values, in this section we focus on unambiguous drawbacks of the off-manifold approach.

### UNCONTROLLED MODEL BEHAVIOUR OFF-MANIFOLD

Sec. 3.1 might lead one to believe that off-manifold Shapley values provide insight into the algebraic dependence of a model. However, the off-manifold approach of evaluating the model on spliced in-distribution data does not constitute a controlled study of such dependence. Off-manifold Shapley values serve as a perilously uncontrolled technique, especially in complex nonlinear models such as neural networks. Indeed, it is widely known that deep-learning models are not robust to distributional shift (Nguyen et al., 2015; Goodfellow et al., 2015). Still, off-manifold Shapley values evaluate the model outside its domain of validity, where it is untrained and potentially wildly misbehaved. This garbage-in-garbage-out problem is the clearest reason to avoid the off-manifold approach.

Since this point has been documented in the literature (Hooker & Mentch, 2019), here we simply provide an example: Fig. 1 shows a binary MNIST digit (LeCun & Cortes, 2010), a coalition of pixels, and 5 random splices that would be used to compute an off-manifold explanation.

### OUTLIER DETECTION EXPLAINED INCORRECTLY OFF-MANIFOLD

To demonstrate that off-manifold Shapley values frequently lead to incorrect explanations, here we offer an example on synthetic data where the ground-truth explanation is known. We generated $10^4$ synthetic data points, each consisting of 20 real-valued features, for the purpose of outlier detection. We split the dataset between 99% inliers and 1% outliers, with the classes generated according to:

$$p_{\text{in}}(x_1, \ldots, x_{20}) = \frac{1}{2} \sum_{z=0,1} \left( \prod_{i=1}^{20} \mathcal{N}[z, \sigma^2](x_i) \right) \tag{11}$$

$$p_{\text{out}}(x_1, \ldots, x_{20}) = \frac{1}{2} \sum_{z=0,1} \left( \prod_{i=1}^{5} \mathcal{N}[\bar{z}, \sigma^2](x_i) \right) \left( \prod_{i=6}^{20} \mathcal{N}[z, \sigma^2](x_i) \right) \tag{12}$$

That is, there is a single binary latent variable $z$. For inliers, each feature is an independent noisy reading of the latent $z$. For outliers, the first 5 features are centred instead around its opposite $\bar{z}$. An example outlier (with $\sigma = 0.05$) is shown in Fig. 3(a). We generated one such data set for each $\sigma \in \{0.01, 0.03, \ldots, 0.15\}$ in order to study the effect of noise on explanation errors.

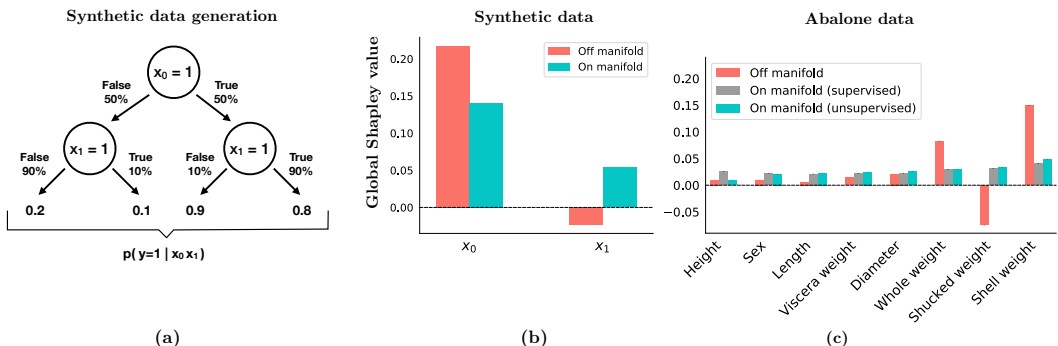

Figure 4: Negative global Shapley values arise off-manifold, in both (a) synthetic and (b) real data.

We fit an isolation forest (Liu et al., 2008) to perform outlier detection on each synthetic dataset, achieving 100% accuracy in every case. We computed the off- and on-manifold value functions of Eqs. (2) and (3) for each isolation forest by sampling the probability distributions directly, as these can be inferred from Eqs. (11) and (12). Figs. 3(b) and 3(c) show the resulting local Shapley values for the example outlier from Fig. 3(a).

The ground-truth explanation of why Fig. 3(a) represents an outlier is that its first 5 features break correlations that exist across 99% of the data. The on-manifold explanation of Fig. 3(c) correctly attributes the 5 largest Shapley values to features $x_1, \ldots, x_5$. The off-manifold explanation of Fig. 3(b) is unambiguously incorrect: feature $x_7$ receives a larger value than $x_2$, $x_4$, and $x_5$. We consider an explanation to be erroneous if $x_1, \ldots, x_5$ do not receive the 5 largest Shapley values.

To show the frequency of incorrect explanations, Fig. 3(d) displays the off- and on-manifold error rates as a function of noise $\sigma$ in the synthetic data set. Incorrect explanations are commonplace off-manifold: one-quarter are in error in the presence of minimal noise, and two-thirds are incorrect at $\sigma = 0.15$. The on-manifold error rate is dramatically lower across this range.

Figs. 3(e) and 3(f) show the root cause of off-manifold errors. These histograms display the distribution of model outputs when evaluated on Shapley coalitions in the off- and on-manifold calculations for $\sigma = 0.05$. In particular, Fig. 3(e) shows the model evaluated on "inlier coalitions" which do not include $x_1, \ldots, x_5$. Note that model outputs for on-manifold coalitions agree with the model evaluated on the actual data, while off-manifold coalitions follow a very different distribution. In particular, since a positive model output indicates a predicted outlier, Fig. 3(e) shows that the off-manifold calculation itself fabricates outliers through its splicing procedure.

Similarly, Fig. 3(f) shows the model evaluated on "outlier coalitions" which do include $x_1, \ldots, x_5$. Note that model outputs are similar for on-manifold coalitions and actual outliers, whereas off-manifold coalitions again differ dramatically. This is a manifestation of uncontrolled model behaviour off the data manifold, and it ultimately leads to erroneous off-manifold explanations.

### BREAKDOWN IN GLOBAL SHAPLEY VALUES OFF-MANIFOLD

To demonstrate that global Shapley values can be misleading off-manifold as well, we generated an additional synthetic data set according to the process in Fig. 4(a). The data has two binary features and a binary label. We fit a decision tree to this data, resulting in a precise match to Fig. 4(a). Note that the features $x_0$ and $x_1$ are positively correlated, both with each other and with label $y$. However, with $x_0$ fixed, the likelihood of $y = 1$ decreases slightly from $x_1 = 0$ to $x_1 = 1$. One might think of $x_0$ as disease severity, $x_1$ as treatment intensity, and $y$ as mortality rate.

Fig. 4(b) displays global Shapley values for this model. The global Shapley values are positive on-manifold, consistent with their interpretation as the portion of model accuracy attributable to each feature. Off-manifold, however, a negative value results from placing too much weight on splices, e.g. with $(x_0, x_1, y) = (0, 1, 1)$, that occur less frequently in the actual data. The negative value would erroneously indicate that $x_1$ is detrimental to the model's overall performance.

We can demonstrate this on real data using UCI Abalone data (Dua & Graff, 2017). We trained a neural network to classify abalone as younger than or older than the median age based on physical

characteristics. Fig. 4(c) displays global Shapley values for this model. (On-manifold values were computed using techniques developed in Sec. 4.1; see App. B for details.)

Observe the drastic difference between the on- and off-manifold explanations in Fig. 4(c). This is due to the tight correlations between features in the data (4 weights and 3 lengths) making the data manifold low-dimensional and important. Notice further the large negative off-manifold global Shapley value, negating its interpretation as the portion of model accuracy attributable to that feature.

# 4 SCALABLE APPROACHES TO ON-MANIFOLD SHAPLEY VALUES

In Sec. 3 we computed on-manifold Shapley values for simple data by estimating $p(x'|x_S)$ from the empirical data distribution or, for synthetic data, by knowing this distribution analytically. Here we introduce two performant methods to compute on-manifold Shapley values on general data. Sec. 4.1 develops the theory underlying our methods, and Sec. 4.2 presents additional experimental results.

## 4.1 THEORETICAL DEVELOPMENT OF ON-MANIFOLD METHODS

Here we develop two methods to learn the on-manifold value function: (i) unsupervised learning the conditional distribution $p(x'|x_S)$, and (ii) a supervised technique to learn the value function directly.

### UNSUPERVISED APPROACH

One can use unsupervised learning to learn the conditional distributions $p(x'|x_S)$ that appear in the on-manifold value function. Here we take an approach similar to Ivanov et al. (2019) to learn these distributions with variational inference. See Douglas et al. (2017) and Belghazi et al. (2019) for alternative techniques to learning conditional distributions that could be used here instead.

Our specific approach includes two model components. The first is a variational autoencoder (Kingma & Welling, 2014; Rezende et al., 2014), with encoder $q_\phi(z|x)$ and decoder $p_\theta(x|z)$. The second is a masked encoder, $r_\psi(z|x_S)$, for which the goal is to map the coalition $x_S$ to a distribution in latent space that agrees with the encoder $q_\phi(z|x)$ as well as possible. A model of $p(x'|x_S)$ is then provided by the composition:

$$\hat{p}(x'|x_S) = \int dz\, p_\theta(x'|z)\, r_\psi(z|x_S) \tag{13}$$

and a good fit to the data should maximise $\hat{p}(x'|x_S)$. A lower bound to its log-likelihood is given by

$$\mathcal{L}_0 = \mathbb{E}_{q_\phi(z|x')}\big[\log p_\theta(x'|z)\big] - \mathcal{D}_{\mathrm{KL}}\big(q_\phi(z|x')\,\|\,r_\psi(z|x_S)\big) \tag{14}$$

While $\mathcal{L}_0$ could be used on its own as the objective function to learn $\hat{p}(x'|x_S)$, this would leave the variational distribution $q_\phi(z|x)$ unconstrained, at odds with our goal of learning a smooth-manifold structure in latent space. This concern can be mitigated by $\mathcal{L}_{\mathrm{reg}} = -\mathcal{D}_{\mathrm{KL}}\big(q_\phi(z|x)\,\|\,p(z)\big)$ which regularises $q_\phi(z|x)$ by penalising differences from a smooth (e.g. unit normal) prior distribution $p(z)$. We thus include $\mathcal{L}_{\mathrm{reg}}$ as a regularisation term in our unsupervised objective: $\mathcal{L} = \mathcal{L}_0 + \beta\,\mathcal{L}_{\mathrm{reg}}$.

### METRIC FOR THE LEARNT VALUE FUNCTION

The unsupervised method presented above leads to a learnt estimate of the conditional distribution, and thus to an estimate of the on-manifold value function: $\hat{v}_{f_y(x)}(S) = \mathbb{E}_{\hat{p}(x'|x_S)}[f_y(x')]$. With the goal of judging the performance of this estimate, consider the following formal quantity:

$$\mathrm{mse}(x_S, y) = \mathbb{E}_{p(x'|x_S)}\big|f_y(x') - \hat{v}_{f_y(x)}(S)\big|^2 \tag{15}$$

This quantity is minimal with respect to $\hat{v}_{f_y(x)}(S)$ when $\hat{v}_{f_y(x)}(S) = \mathbb{E}_{p(x'|x_S)}[f_y(x')]$, in agreement with the definition, Eq. (3), of the on-manifold value function. We can then quantitatively judge the performance of the unsupervised model $\hat{p}(x'|x_S)$ by computing

$$\mathrm{MSE} = \mathbb{E}_{p(x)}\, \mathbb{E}_{S\sim\mathrm{Shapley}}\, \mathbb{E}_{y\sim\mathrm{Unif}}\big|f_y(x) - \hat{v}_{f_y(x)}(S)\big|^2 \tag{16}$$

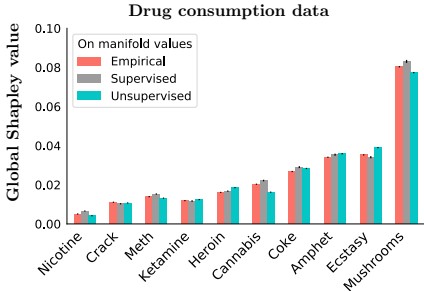

Figure 5: Validation of unsupervised and supervised techniques for computing on-manifold Shapley values. Comparison against empirical ground truth, which appeared as "On manifold" in Fig. 2(b).

Table 1: Performance and stability, in terms of MSE, of supervised and unsupervised approaches. Performance is compared with off-manifold splicing and, where accessible, the empirical optimum.

| DATA SET | OFF MANIFOLD | UNSUPERVISED | SUPERVISED | EMPIRICAL |
|----------|-------------|--------------|------------|-----------|
| DRUG | 0.0634 | $0.0536 \pm 0.0007$ | $0.0441 \pm 0.0002$ | 0.0436 |
| ABALONE | 0.0647 | $0.0293 \pm 0.0009$ | $0.0200 \pm 0.0001$ | – |
| CENSUS | 0.0344 | $0.0300 \pm 0.0006$ | $0.0250 \pm 0.0001$ | – |
| MNIST | 0.0448 | $0.0257 \pm 0.0005$ | $0.0121 \pm 0.0001$ | – |

Note that this is precisely Eq. (15) averaged over coalitions $S$ drawn from the Shapley sum,[2] features $x_S \sim p(x_S)$ drawn from the data, and labels $y$ drawn uniformly over classes. Moreover, the mean-square-error in Eq. (16) is easy to estimate using the empirical distribution $p(x)$ and the learnt model $\hat{p}(x'|x_S)$, thus providing an unambiguous metric to judge the outcome of the unsupervised approach.

SUPERVISED APPROACH

The MSE metric of Eq. (16) supports a supervised approach to learning the on-manifold value function directly: one can define a surrogate model $g_y(x_S)$ that operates on coalitions of features $x_S$ (e.g. by masking out-of-coalition features) and that is trained to minimise the loss:

$$\mathcal{L} = \mathbb{E}_{p(x)} \, \mathbb{E}_{S \sim \text{Shapley}} \, \mathbb{E}_{y \sim \text{Unif}} \left| f_y(x) - g_y(x_S) \right|^2 \tag{17}$$

As discussed above Eq. (16), this loss is minimised as the surrogate model $g_y(x_S)$ approaches the on-manifold value function $\mathbb{E}_{p(x'|x_S)}[f_y(x')]$ of the model-to-be-explained.

## 4.2 ADDITIONAL EXPERIMENTS

Sec. 3 above presented two experiments using the scalable on-manifold methods developed here. In particular, Fig. 2(a) applied the unsupervised method to Census Income data, showing that on-manifold Shapley values detect hidden model dependence on sensitive features, and Fig. 4(c) applied both methods to Abalone data, showing that global Shapley values remain positive and interpretable on-manifold. In this section, we perform additional experiments to study the performance and stability of Sec. 4.1's methods, as well as their effectiveness on higher-dimensional data.

PERFORMANCE AND STABILITY

Our implementations of the unsupervised and supervised approaches to on-manifold Shapley values are summarised in Apps. A and B. Both approaches lead to broadly similar results. Fig. 5 compares the two techniques on the Drug Consumption data, where explanations are compared against the ground-truth empirical computation from Fig. 2(b).

The unsupervised approach is flexible but untargeted: $p(x'|x_S)$ is data-specific but model-agnostic, accommodating explanations for many models trained on the same data. The supervised approach

---

[2]In more detail, here we sample coalitions from Eq. (1), where the probability assigned to each coalition is the combinatorial factor $|S|!(n - |S| - 1)!/n!$.

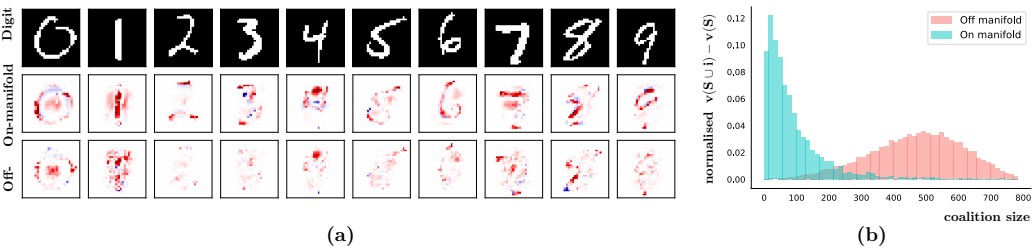

(a)                                                          (b)

Figure 6:   (a) Randomly drawn MNIST digits explained on / off manifold.  Red / blue pixels indicate positive / negative Shapley values, and the colour scale in each column is fixed.  (b) Shapley summand as a function of coalition size – averaged over coalitions, pixels, and the MNIST test set.

must be retrained on each model, but it entails direct minimisation of the MSE. The supervised method is thus expected to achieve higher accuracy. We confirmed this on all data sets studied in this paper; see Table 1 for a numerical comparison of the MSEs.

In Table 1, central values indicate the test-set MSE achieved by each method. The table compares the unsupervised and supervised methods against off-manifold splicing, showing significant improvement over this baseline. Note that an MSE of zero is not achievable, because $f_y(x)$ in Eq. (16) or (17) is not fully determined by partial input $x_S$. For the Drug Consumption data where we can compute $p(x'|x_S)$ empirically, the optimal MSE happens to be $0.0436$.

Uncertainties in Table 1 represent the standard deviation in test-set MSE upon repeating each method with fixed hyperparameters 10 times. (Uncertainties are absent for the off-manifold and empirical columns, as these do not involve training a separate model.) The table thus indicates that the supervised method offers increased stability as compared to the unsupervised approach.

The supervised method is more efficient as well: while the unsupervised technique estimates the value function by sampling from $\hat{p}(x'|x_S)$, the supervised approach learns the value function directly. The supervised method thus requires far fewer model evaluations to match the standard-error of the unsupervised method: roughly 10 times fewer in our experiments.

EXAMPLE ON MNIST

To demonstrate on-manifold explainability on higher-dimensional data, we trained a fully connected network on binary MNIST (LeCun & Cortes, 2010) and explained random digits in Fig. 6(a).

Despite having the same sum over pixels – as controlled by the local version of Eq. (5) – and explaining the same model prediction, each on-manifold explanation is more concentrated, with more interpretable structure, than its off-manifold counterpart. The handwritten strokes are clearly visible on-manifold, with key off-stroke regions highlighted as well. Off-manifold explanations generally display lower intensities spread less informatively across the digit-region.

These off-manifold explanations are a result of splices as in Fig. 1. With such unrealistic input, the model's output is uncontrolled and less informative. In fact, it is only on very large coalitions of pixels, subject to minimal splicing, that the model can make intelligent predictions off-manifold. This is confirmed in Fig. 6(b), which shows the average Shapley summand as a function of coalition size on MNIST. Note that primarily large coalitions underpin off-manifold explanations, whereas far fewer pixels are required on-manifold, consistent with the low-dimensional manifold underlying the data.

## 5   CONCLUSION

In this work, we took a careful study of the off-manifold problem in AI explainability. We presented important distinctions between on- and off-manifold explainability and provided experimental evidence for several novel shortcomings of the off-manifold approach. We then introduced two techniques to compute on-manifold Shapley values on general data: one technique learns to impute features on the data manifold, while the other learns the Shapley value-function directly. In-so-doing, we provided compelling evidence against the use of off-manifold explainability, and demonstrated that on-manifold Shapley values offer a viable approach to AI explainability in real-world contexts.

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

## A    IMPLEMENTATION DETAILS

For the unsupervised approach, we modelled the encoder $q_\phi(z|x)$ as a diagonal normal distribution with mean and variance determined by a neural network:

$$q_\phi(z|x) = \mathcal{N}\big(\mu_\phi(x), \sigma_\phi(x)\big) \tag{18}$$

We modelled the decoder $p_\theta(x|z)$ as a product distribution:

$$p_\theta(x|z) = \prod_i p_\theta(x_i|z) \tag{19}$$

where the distribution type (e.g. normal, categorical) of each $x_i$ is chosen per-data-set and each distribution's parameters are determined by a shared neural network. We modelled the masked encoder $r_\psi(z|x_S)$ as a Gaussian mixture:

$$r_\psi(z|x_S) = \sum_j w_\phi^{(j)}(x)\, \mathcal{N}\Big(\mu_\phi^{(j)}(x), \sigma_\phi^{(j)}(x)\Big) \tag{20}$$

To allow $r_\psi(z|x_S)$ to accept variable-size coalitions $x_S$ as input, we simply masked out-of-coalition features with a special value $(-1)$ that never appears in the data.

The unsupervised method has several hyperparameters: $\beta$ which multiplies the regularisation term, the number of components in Eq. (20), as well the architecture and optimisation of the networks involved. For each experiment in this paper, we tuned hyperparameters to minimise the MSE of Eq. (16) on a held-out validation set; see App. B for numerical details.

For the supervised approach, we modelled $g_y(x_S)$ using a neural network, again masking out-of-coalition features (with $-1$) to accommodate variable-size coalitions $x_S$. This method's hyperparameters, relating to architecture and optimisation, were similarly tuned to minimise the validation-set MSE; see App. B for details.

## B    DETAILS OF EXPERIMENTS

Here we provide numerical details for the experiments presented in the paper.

### B.1    DRUG CONSUMPTION EXPERIMENT

On the Drug Consumption data from the UCI repository (Dua & Graff, 2017), we used 10 binary features from the data set – Mushrooms, Ecstasy, etc., as displayed in Fig. 5 – to predict whether individuals had ever consumed an 11th drug: LSD. The explanations of Fig. 2(b) and Fig. 5 describe a random forest fit with default sklearn parameters and max_features = None, which achieves 82.2% test-set accuracy amidst a $57 : 43$ class balance.

In Fig. 2(b), global off-manifold Shapley values were computed using $10^6$ Monte Carlo samples of Eq. (4). For each labelled data point $(x, y)$ sampled from the test set, a single permutation was drawn to estimate Eq. (1) and a single data point $x'$ was drawn to estimate the off-manifold value function Eq. (2). In all the figures of this paper, bar height represents the mean that resulted from Monte Carlo sampling, and error bars display the standard error of the mean.

Global on-manifold Shapley values in Fig. 2(b) were computed similarly, but in this case using the on-manifold value function of Eq. (3). For each sampled coalition $x_S$, a random data point $x'$ was drawn from the test set, with the crucial requirement that $x'_S = x_S$. In the text, we refer to this as empirically estimating the conditional distribution $p(x'|x_S)$. Such empirical estimation is only possible because this data set has a small number of all-binary features.

Tree SHAP values in Fig. 2(b) were computed with the SHAP package (Lundberg & Lee, 2017) with model_output = margin and feature_perturbation = tree_path_dependent.

The values labelled "Model retraining" in Fig. 2(b) were computed by fitting a separate random forest $g_S$ for each coalition $S$ of features in the data set: $2^{10}$ models in all. We used these models to compute the sum of Eq. (10), where $A(g_S)$ represents a variant of model $g_S$'s accuracy: it is the

Table 2: Optimal hyperparameters found for computing on-manifold Shapley values.

| DATA SET | METHOD | HIDDEN DIM. | LEARN. RATE | LATENT DIM. | MODES | $\beta$ |
|---|---|---|---|---|---|---|
| DRUG | SUPERVISED | 512 | $10^{-3}$ | | | |
| | UNSUPERVISED | 128 | $10^{-3}$ | 4 | 1 | 0.5 |
| ABALONE | SUPERVISED | 512 | $10^{-3}$ | | | |
| | UNSUPERVISED | 256 | $10^{-3}$ | 2 | 1 | 0.05 |
| CENSUS | SUPERVISED | 512 | $10^{-3}$ | | | |
| | UNSUPERVISED | 128 | $10^{-3}$ | 8 | 1 | 1 |
| MNIST | SUPERVISED | 512 | $10^{-4}$ | | | |
| | UNSUPERVISED | 512 | $10^{-4}$ | 16 | 1 | 1 |

accuracy achieved if one predicts labels by drawing stochastically from $g_S$'s predicted probability distribution (as opposed to deterministically drawing the maximum-probability class).

The global on-manifold Shapley values in Fig. 2(b) appear in Fig. 5 as well, labelled "Empirical". Fig. 5 also displays on-manifold Shapley values computed using the supervised and unsupervised methods introduced in this paper. As above, these are Monte Carlo estimates of Eq. (4). The supervised method involved training a fully connected network on the MSE loss of Eq. (17). All neural networks in this paper used 2 flat hidden layers, Adam (Kingma & Ba, 2015) for optimisation, and a batch size of 256. We scanned over a grid with

$$\text{hidden layer size} = \{128,\ 256,\ 512\} \tag{21}$$
$$\text{learning rate} = \{10^{-3}, 10^{-4}\}$$

choosing the point with minimal MSE on a held-out validation set after 10k epochs of training; see Table 2. Each supervised value in Fig. 5 corresponds to $10^4$ Monte Carlo samples.

The unsupervised method involved training a variational autoencoder as described in Sec. 4.1 and App. A. The encoder, decoder, and masked encoder were each modelled using fully connected networks, trained using early stopping with patience 100. We scanned over a grid of hidden layer sizes and learning rates as in Eq. (21) as well as

$$\text{latent dimension} = \{2,\ 4,\ 8,\ 16\} \tag{22}$$
$$\text{latent modes} = \{1, 2\}$$
$$\text{regularisation } \beta = \{0.05,\ 0.1,\ 0.5,\ 1\}$$

choosing the point with minimal validation-set MSE; see Table 2. Unsupervised values in Fig. 5 correspond to $10^6$ Monte Carlo samples.

## B.2 CENSUS INCOME EXPERIMENT

To produce the explanations of Fig. 2(a) we used the Census Income data set from the UCI repository (Dua & Graff, 2017). The data contains 49k individuals from the 1994 US Census, as well as 13 features which we used to predict whether annual income exceeded \$50k. We trained a fully connected network (hidden layer size 50, default sklearn parameters, and early stopping), achieving a test-set accuracy of 85% amidst a 76 : 24 class balance.

The Shapley values for this model are labelled "Original model" in Fig. 2(a). These were computed exactly as described in App. B.1, except that the supervised method used 5k epochs, and the unsupervised method used patience 50. Optimised hyperparameters are given in Table 2. The on-manifold values in Fig. 2(a) were computed using the unsupervised method. While the supervised method does not appear in the figure, it was performed to complete Table 1.

We also fine-tuned the "Original model" to suppress the importance of sex. Motivated by Dimanov et al. (2020) we added a term to the loss that penalises the finite difference in the model output with

respect to sex (as this is a discrete feature). The modified loss function thus becomes

$$\frac{1}{N} \sum_{i=1}^{N} \mathcal{L}\big(f(x_i), y_i\big) \; + \; \alpha \left| f\big(x_i \,|\, \mathrm{do}(\mathrm{sex} = 1)\big) - f\big(x_i \,|\, \mathrm{do}(\mathrm{sex} = 0)\big) \right| \qquad (23)$$

where $\mathcal{L}$ is the cross-entropy loss, $f\big(x_i \,|\, \mathrm{do}(\mathrm{sex} = j)\big)$ denotes $f$ evaluated on the data point $x_i$ with the value for sex replaced with $j$, and $\alpha$ is a hyperparameter controlling the trade-off between optimising the accuracy and minimising the effect of sex. We fine-tuned the model for an additional 200 epochs with $\alpha = 3$. The resulting model agrees with the baseline on over 98.5% of the data, and has the same test-set accuracy. Shapley values for this model are labelled "Suppressed model" in Fig. 2(a).

### B.3    ABALONE EXPERIMENT

The Abalone data set from the UCI repository (Dua & Graff, 2017) contains 8 features corresponding to physical measurements (see Fig. 4c) which we used to classify abalone as younger than or older than the median age. We trained a neural network to perform this task – with hidden layer size 100, default sklearn parameters, and early stopping – obtaining a test-set accuracy of 78%.

Shapley values in Fig. 4(c) were computed exactly as described in App. B.1, except that the supervised method involved training for 5k epochs. Optimised hyperparameters are given in Table 2.

### B.4    MNIST EXPERIMENT

For binary MNIST (LeCun & Cortes, 2010), we trained a fully connected network (hidden layer size 512, default sklearn parameters, and early stopping) achieving 98% test-set accuracy.

The digits in Fig. 6(a) were randomly drawn from the test set. Shapley values in Fig. 6(a) were computed exactly as described in App. B.1, except that the supervised method involved training for 2k epochs, and the on-manifold explanations are based on 16k Monte Carlo samples per pixel. Optimised hyperparameters are given in Table 2. The on-manifold explanations in Fig. 6(a) were computed using the supervised method. While the unsupervised method does not appear in the figure, it was performed to complete Table 1.

The average uncertainty, which is not shown in Fig. 6(a), is roughly 0.002 – stated as a fraction of the maximum Shapley value in each image.

