# OpenReview forum: "Shapley explainability on the data manifold"
_ICLR.cc/2021/Conference — ICLR 2021 Poster_

### Official Review · AnonReviewer2 · 2020-10-25
**The use of conditional generative model seems to be reasonable.**

**Rating:** 6
**Confidence:** 3

**Review:**

### Paper Summary
In this paper, the authors pointed out that the explanation with Shapley value can be misleading due to inappropriate assumptions in the current estimation method. The authors pointed out that the "splicing" techniques used in the current estimation method assumes the variables to be independent, which is not always the case in practice.  Application of "splicing" to dependent variables can yield unrealistic data that is outside the data manifold. The authors demonstrated that the use of such unrealistic data outside the data manifold lead to inappropriate estimate of Shapley value (Fig. 2, 3, 4, 5). To mitigate the effect of such unrealistic data, the authors proposed using generative models to sample realistic data inside the data manifold, thus avoiding use of unrealistic data for estimating Shapley value.

### Quality & Clarity
I think the paper is written clearly, and the main claims are easy to follow.
The experiments are designed well to support the claim.

### Originality & Significance
#### Contribution of the paper
Essentially, this paper tackles the known problem (avoiding unrealistic data in Shapley value (*1)) by using a common technique (generative model to sample realistic data (*2)). Although the idea is straightforward, the method itself seems to be reasonable and would be useful in some extent.

(*1) The problem considered in this paper is not a new problem. The problem of using unrealistic data for estimating Shapley value is pointed out by several previous studies [Ref1, 2, 3, 4], as some of them are mentioned in the paper.

(*2) The major contribution of this study is therefore on the methods for mitigating the effect of unrealistic data. In this study, the authors proposed using generative models to sample realistic data inside the data manifold. The authors also proposed training a surrogate model $g_y$ that approximates the conditional expectation used for computing Shapley value. To my knowledge, such techniques were not considered in the previous studies.  However, the use of generative models for sampling from conditional distribution itself is a common technique, e.g. [Ref5,6] used GAN and VAE for Model-X Knockoff.

#### Explanation and data manifold
The relationship between the data manifold and explanation is studied by [Ref7,8], which is missing in the current paper. It would be great if there are some discussions how the current paper and [Ref7, 8] are similar/different.
Below, I list some of them.
* Similar Points
  The similar problem, the explanation outside the data manifold, are studied in [Ref7, 8].
  The use of autoencoder is considered to mitigate the effect of data outside the data manifold.
* Different Points
  [Ref7, 8] focus on the gradient-based methods, while the current paper focuses on Shapley value.
In addition to similarity/difference, I am also interested in whether the theories of [Ref7, 8] are applicable to Shapley values, or whether there is a fundamental differences between the gradient-based methods and Shapley value.

[References]
* [Ref1] Problems with Shapley-value-based explanations as feature importance measures, ICML20
* [Ref2] The many Shapley values for model explanation, ICML20
* [Ref3] Explaining black box decisions by Shapley cohort refinement, arXiv19
* [Ref4] Interpretable Machine Learning (Sec 5.10.2), 2019
* [Ref5] KnockoffGAN: Generating Knockoffs for Feature Selection using Generative Adversarial Networks, ICLR19
* [Ref6] Deep latent variable models for generating knockoffs, 2020
* [Ref7] Explanations can be manipulated and geometry is to blame, NeurIPS19
* [Ref8] Fairwashing Explanations with Off-Manifold Detergent, ICML20

### Pros & Cons
[Pros]
* The use of generative models for sampling realistic data from conditional distribution seems to be reasonable to mitigate the effect of unrealistic data used in the current method "splicing."

[Cons]
* The use of generative models for sampling from conditional distribution itself is a common technique.
* The relationship between the data manifold and explanation is studied by [Ref7,8], which is missing in the current paper. It would be good to provide some discussions how the current paper and [Ref7, 8] are similar/different.

---
I read the rebuttal and the updated version of the paper. I think the authors did a good job resolving my concerns on novelty. I updated my score.

---

> ### Author Response · Authors · 2020-11-17
> **A note on our contributions and a clarification of related literature**
>
> We appreciate the reviewer’s comments on the quality and clarity of our work and are grateful for their thoughts on the related literature.
>
>
> Responding to the reviewer’s starred comments:
>
> **Point (1)** While we agree that the off-manifold problem in explainability is already widely acknowledged in the literature, we consider Sec. 3 of our paper to be a _significant secondary contribution_. Indeed, Sec. 3 presents several novel and precise shortcomings of off-manifold explainability that have yet to appear in the literature. These include:
> * Off-manifold explanations can have a much higher error rate than their on-manifold counterparts. This is especially true in an application like outlier detection, where the data manifold is central to the modelling task.
> * Global Shapley values can be negative off-manifold, in conflict with their interpretation as the portion of model accuracy attributable to each input feature.
> * On-manifold Shapley values approximate “model retraining Shapley values” (i.e. Eq 10 in our paper), while their off-manifold counterparts do not.
>
> **Point (2)** Our “unsupervised method” is meant as a proof of concept that one can use generative modelling to practically and effectively compute on-manifold Shapley values. We do not have strong opinions on which specific generative technique should be used to learn the requisite conditional probability distributions, and this will certainly vary across applications. We will update the manuscript to reflect this position.
>
> However, we do not believe Refs. [5, 6] provide methods to learn the conditional probability distributions required for on-manifold explainability. From reading [5], it is our understanding that the “knockoff” framework generates a realistic new data point $\tilde X$ that is independent of the label $Y$ given a true data point $X$, whereas what we require is samples from $p(x_{\bar S}|x_S)$ for any possible coalition $S$.
>
>
> In addition, we thank the reviewer for pointing out Refs [7, 8], which we missed. We will update our paper to mention each of these relevant works. We found [8] to be especially interesting and will discuss it in our Sec 1 and Sec 3.1. Here are our thoughts in more detail:
>
> **(Ref 7)** This paper relates to our Sec 3.1 (“Hidden model dependence on sensitive attributes”). The paper shows that (for gradient-based rather than Shapley-based explanation methods) input features can be perturbed to change the model’s explanation without significantly changing its output. One major difference with our work is that [7] considers varying the input data point (a la adversarial examples) whereas our Sec 3.1 considers varying the model. As such, there are different root causes for the misleading explanations in each case. In [7] the cause is uncontrolled curvature in the model output, and the solution is a $\beta$ parameter to smooth the model’s softplus nonlinearities. In our Sec 3.1, the cause is off-manifold model evaluations, and we solve it with on-manifold explainability.
>
> **(Ref 8)** This paper essentially considers the off-manifold problem as it manifests in gradient-based model explainability. The fundamental realisation here is that two models can agree on the data manifold but have very different gradients in directions orthogonal to the data manifold, thus leading to different gradient-based explanations for the two models. The solution presented in [8] is to instead take _directional derivatives_ along the data manifold to compute the explanation, and these directional derivatives are essentially found by differentiating an autoencoder’s decoder’s output with respect to its latent representation.
>
> The main difference between [8] and our work is that the off-manifold problem manifests differently in Shapley-based explainability and thus must be solved differently as well:
> * In off-manifold Shapley explainability, the model is generically evaluated far from the data manifold. (Whereas in gradient-based explainability, the model's gradients are evaluated _on_ the data manifold – just potentially in off-manifold directions.)
> * To solve the off-manifold problem, one must learn the conditional probability distributions of the data. These are not learnt by an autoencoder (or VAE) and require a more involved approach, such as those we presented in our paper.
>
>
> Having thus responded to all the reviewer’s comments, we hope the reviewer will consider a more positive scoring of our work.
>
>
> [5] KnockoffGAN: generating knockoffs for feature selection using generative adversarial networks, ICLR 2019.
> [6] Deep latent variable models for generating knockoffs, 2020.
> [7] Explanations can be manipulated and geometry is to blame, NeurIPS 2019.
> [8] Fairwashing explanations with off-Manifold detergent, ICML 2020.

---

### Official Review · AnonReviewer3 · 2020-10-26
**In this well-designed and motivated study the authors highlight shortcoming of existing methods for computing Shapley values. They propose two methods for mitigation that enable sampling form estimate on-manifold conditionals. The methods are shown to be effective.**

**Rating:** 8
**Confidence:** 3

**Review:**

The paper gives thorough intuitions, theoretical arguments and synthetic examples that demonstrate the shortcomings of off-manifold Shapley values. It then proposes two approaches to overcome this limitation. Specifically, they enable scalable sampling from the on-manifold conditionals. The first approach is unsupervised based on VAEs and model-agnostic. The second approach builds upon the supervised model. The latter is shown to be more accurate and data efficient. Both approaches are shown to fix the limitations of off-manifold Shapley values on a real-world MNIST example. The authors provide implementation detail that will be useful for practitioners.
Thanks for this a well-written, well-motivated easy-to-follow paper.

Minor comments.
- add to the intro a defintion of "data splicing"
- formalize problem of off-manifold shapley values
- It would be great if the authors released code for their experiments.

---

> ### Author Response · Authors · 2020-11-17
> **We appreciate the summary of our work and the positive review**
>
> We are happy to hear that the reviewer found our paper to be well-written and well-motivated. If the reviewer could clarify what they mean by “formalize the problem of off-manifold Shapley values” we would be eager to further improve our paper according to this feedback. Thank you!

---

### Official Review · AnonReviewer4 · 2020-10-28
**New approach for calculating SHAP values while holding out features with the conditional distribution**

**Rating:** 7
**Confidence:** 5

**Review:**

This paper explores the drawbacks of existing explanation methods that implicitly use off-manifold samples while holding out groups of features. Most existing SHAP variants (e.g., KernelSHAP) use off-manifold samples, in part because using on-manifold samples from the conditional distribution is computationally challenging. This work proposes two methods to do so tractably (one "unsupervised," one "supervised") and demonstrates some advantages of the on-manifold approach.

I appreciated many results from this paper. It was valuable to explain that when using the on-manifold approach, two models that make identical predictions get identical SHAP values. It was also good to see that the on-manifold approach successfully detects dependencies on sensitive attributes where the off-manifold approach fails (Figure 2). When comparing the various approaches, showing that the on-manifold approach is roughly equivalent to retraining $2^n$ models is helpful, and showing that TreeSHAP does not perfectly model the conditional distribution corrects a potential misconception that this approach is equivalent to the on-manifold approach.

A couple of questions and concerns about the methods and theory in this work.

1. The perspective of the data distribution being controlled by unobserved latent variables and noise variables was not helpful, in my view. It was not clear what the authors meant by $O(\epsilon)$, it was not clear why we should assume that the relationship between $x$ and $z$ is invertible (see $\Psi_k$) and it was not clear when/why we should assume that each $x_i$ is a deterministic function of the other features (see $\tilde \Psi$). The math in this section (beginning of 3.1) seemed like another way of saying that data samples should come from dense parts of the data distribution, but without adding much additional insight. (I'm open to changing my opinion on this if others disagree.)
2. The metric for the on-manifold value function was not explained clearly. In what sense is this value function optimal? Is it necessary to use MSE to train a supervised classification model? I believe a more complete proof of this result is provided in Appendix A of Covert et al. (cited), and this result suggests how to train a supervised surrogate using cross entropy loss.
3. The comparison of the supervised and unsupervised approach is very important because it informs how future work should use the on-manifold method. I understand the space constraints, but if possible some of these results could be shown in the main text. Table 1 in the supplement seems to show that the supervised approach is vastly superior according to the proposed metric; the authors should also include a comparison with the off-manifold approach (using the marginal distribution) to give a sense of how significant the difference is. It would also be interesting to see the standard error comparison mentioned on p.8, which suggests that the supervised approach converges much faster.

A couple presentation concerns:

- Removing features using their conditional distribution is more specific than using on-manifold samples, yet this is what the authors seem to mean by "on-manifold." (It is possible to use samples from a different distribution that shares the same support as the conditional distribution.) The authors may wish to clarify this ambiguity and possibly explain more specifically why the conditional distribution is appropriate.
- The presentation is often specific to classification models, but this seems unnecessary. Readers can probably translate the ideas to regression models on their own, but the authors may consider broadening their presentation.
- The "non-parametric limit" is an odd way of presenting the result in Eq. 9 (in my view, although others may disagree). In what sense is the model converging? To be more clear, the authors may consider explaining that this result requires the optimal model (i.e., the Bayes classifier) and that this should only be expected given unlimited data and a sufficiently flexible model.
- When discussing the "unsupervised approach," it may be worth saying explicitly that this does not directly provide an estimate of the value function, but it provides the means to estimate it via a Monte Carlo approximation. The authors may also wish to point out that this means the unsupervised approach requires far more model evaluations than the supervised approach to evaluate any given coalition.
- What exactly is the Shapley distribution used in Eq. 16? I couldn't find where this was explained.

The paper has a couple missing citations, although they don't particularly diminish the work's novelty. Slack et al. [3] show that off-manifold methods can be fooled into missing dependencies on sensitive attributes; this could fit into the narrative in a couple parts of the paper. The "unsupervised approach" is just learning to sample from arbitrary conditional distributions, and there's more work on this topic than the Ivanov et al. paper. Douglas et al. [2] have a method called the "universal marginalizer" and Belghazi et al. [1] have a method called the "neural conditioner" (NC) that has been shown to outperform the AC-VAE. The authors may wish to comment why their approach is modeled off of the AC-VAE rather than the more recent NC.

Overall, I think that this work proposes a solution to an important problem (holding out features using their conditional distribution) and I believe it could be impactful. However, I also think the paper's presentation could be improved in the ways described above.

[1] Belghazi et al., "Learning about an exponential amount of conditional distributions" (2019)

[2] Douglas et al., "A Universal Marginalizer for Amortized Inference in Generative Models" (2017)

[3] Slack et al., "Fooling LIME and SHAP: Adversarial Attacks on Post hoc Explanation Methods" (2019)

##########
Update
##########

The authors have addressed many of my concerns. Moving Table 1 into the main text and adding the new columns is an important change, and it seems that the authors have improved various aspects of their presentation. I'm raising my score to a 7.

---

> ### Author Response · Authors · 2020-11-17
> **Several improvements to our presentation**
>
> We thank the reviewer for a very thorough consideration of our work.
>
> We first respond to the reviewer’s comments on our theory and methodology:
>
> **Point 1:** We appreciate the reviewer’s perspective on our formal presentation of the data manifold in Sec. 3.1. We included this to build intuition for a lower dimensional data manifold in ambient higher dimensional space. It also provides a simple way (through Eq. 8) of understanding how models could have different algebraic dependence while maintaining identical behaviour on the data. To respond to the reviewer’s specific questions:
> * By $O(\epsilon)$ we just mean some function of $\epsilon$ that goes to zero as $\epsilon \to 0$. We could replace this with simply “$0$ as $\epsilon \to 0$” in Eq. 7 if this would be clearer.
> * The assumption, that the relationship between $x$ and $z$ is invertible, is without loss of generality. Indeed, if the data lives on a $d$-dimensional manifold in $n$-dimensional space, then the existence of $n - d$ constraint equations is a tautology. We will update the manuscript to simplify the presentation here and sidestep this assumption.
> * The existence of a constraint equation among the features means that $x_1$, for example, implicitly depends on the other $x_i$’s.
>
> **Point 2:** We use the MSE to train the supervised surrogate model because this results in a surrogate model that outputs the average of the original model’s output, conditioned on the in-coalition features. This is what is required by the on-manifold value function of Eq. 3.
>
> **Point 3:** As our submission is now allowed to utilise a 9th page, we will move Table 1 into the main text as the reviewer suggested.
>
> Next we respond to the reviewer’s comments on our presentation:
> * We agree that, out of context, the term “on manifold” could correspond to any distribution with the same support as the data distribution. However, we feel that our presentation makes it sufficiently clear that we use “on manifold” to describe the value function defined with respect to the data’s conditional distribution.
> * We will add a sentence in Sec. 2 stating that the results of our paper can be applied to regression problems by reinterpreting $f_y(x)$ as the model’s predicted value rather than its predicted probability.
> * We will change the term “non-parametric limit” to “optimal-model limit” and clarify that one approaches this in the limit of abundant data and high model capacity.
> * We believe our paper already implies the points suggested by the reviewer. See the final paragraph in Sec 4.2 (“Performance and stability”). We will revise it slightly to say: “...while the unsupervised technique estimates the value function by sampling from $p(x’ | x_S)$, the supervised approach learns the value function directly. The supervised method thus requires far fewer model evaluations to match the standard-error of the unsupervised method: roughly 10 times fewer in our experiments.”
> * The “Shapley distribution” referenced in Eq. 16 is the distribution over coalitions implied by Eq. 1, where the probability assigned to each coalition is the combinatorial factor $|S|! (n - |S| - 1)! / n!$. We will clarify this under Eq. 16.
>
> We thank the reviewer for pointing out additional relevant citations. We will cite [3] in Sec 3.1 and [1, 2] in Sec 4.1. Our “unsupervised method” is meant as a proof of concept that one can use generative modelling to practically and effectively compute on-manifold Shapley values. We do not have strong opinions on which specific generative technique should be used to learn the requisite conditional probability distributions, and this will presumably vary across applications. We will update the manuscript to reflect this position.
>
> We agree with the reviewer that our paper tackles an important problem and thus has potential for impact. Having made the above improvements to our paper’s presentation, we hope the reviewer will consider an increased scoring of our work.
>
> [1] Belghazi et al., "Learning about an exponential amount of conditional distributions" (2019)
> [2] Douglas et al., "A Universal Marginalizer for Amortized Inference in Generative Models" (2017)
> [3] Slack et al., "Fooling LIME and SHAP: Adversarial Attacks on Post hoc Explanation Methods" (2019)

---

> ### Author Response · Authors · 2020-11-20
> **Off-manifold entries added to Table 1**
>
> As noted in our general comment above, we thank the reviewer for the excellent suggestion of adding an "Off manifold" column to Table 1. We've updated our paper with this addition, and we've added corresponding discussion to the top of page 9.

---

### Official Review · AnonReviewer1 · 2020-10-29

**Rating:** 7
**Confidence:** 4

**Review:**

This paper is focused on the off-data manifold problem with Shapley values which is created by sampling data that is out of distribution. The goal is to develop efficient methods. Two main algorithms are proposed: Generative models to approximate conditional distributions and training supervised models for direct approximation. They show an advantage against original off-manifold Shapley values in experiments. The impression experiment is particularly interesting.
The overall idea of using generative models to solve the issue of off-manifold data in interpretability methods (not just SHAP) is generally a nice direction. Note that the problem is more prominent for the case of SHAP as the method is built on performance on all subsets of the input features and the paper makes a good case of showing the necessity of solving the off-manifold data problem for Shapley-based methods. The experimental results are also comprehensive and provide enough evidence for their usefulness.
 There seem to be novelty concerns "A performant method to estimate onmanifold Shapley values on general data is until-now lacking and a focus of this work." The problem of off-manifold data in interpretability is well studied and its specifics for the SHAP method have been discussed before https://proceedings.icml.cc/static/paper_files/icml/2020/334-Paper.pdf The authors make a very brief reference to this paper but do not actually mention how they are different. Unless the main differential contribution of the work to the existing literature is clear, I cannot change my score. If the contribution is ""experimental evidence ... in favour of the onmanifold approach", the contribution is not enough for this venue.

Questions and notes:
* The higher dimensional example is not actually high dimension and it does not give a sense of how the method would be applied to cases like ImageNet. Authors need to provide runtime results to make the case for the practical efficiency of their method.
* There does not seem to be a discussion of the effect of the generative model's performance on the results. Having a good generative model that actually captures conditional probabilities for high dimensional data like images is still an open question and therefore, it will hinder the proposed method's performance.

---

> ### Author Response · Authors · 2020-11-12
> **Differential contribution relative to [1] clarified**
>
> We appreciate the reviewer’s thoughtful summary of our paper as well as their positive comments.
>
> The reviewer’s primary concern is the novelty of our work. In particular, the reviewer wonders what differentiating contribution our paper makes on top of previous work in [1]. We see our contributions as quite dissimilar from those of [1]. Put succinctly, [1] acknowledges the off-manifold problem in explainability but does not solve it robustly. In contrast, our paper introduces and benchmarks two _general solutions_ for using deep learning to compute on-manifold Shapley values for complex data. Indeed, the closing sentence of [1] refers to how a robust solution to on-manifold Shapley values is still lacking: “The remedy to this common problem requires modelling the true feature distribution…”; this is what we address in our paper. In more detail:
> * Sec 3 of [1] outlines a procedure for computing “Conditional Expectation Shapley” values (what we would refer to as on-manifold Shapley values) using an empirical estimate of the data’s conditional probability distributions. However, [1] clearly acknowledges that this is not a practical algorithm, stating that it is “extremely sensitive to the degree of sparsity; sparsity arises naturally when the variables are continuous” (or high-dimensional). In the case of continuous features, the paper suggests making a fairly strong smoothness assumption, drawing samples from the data set that have feature values in the neighbourhood of (instead of equal to) the correct conditional values. However, this approach suffers from the curse of dimensionality; moreover, it does not work for categorical features.
> * As our major differentiating contribution, our work introduces a general and robust solution to computing on-manifold explanations by _learning_ the requisite conditional probability distributions.
> * It’s also worth mentioning that the other methods discussed in [1], i.e. “Baseline Shapley” and “Integrated Gradients”, are unambiguously off-manifold methods. Baseline Shapley involves setting all out-of-coalition features to a fixed value. Integrated Gradients involves performing a line-integral over each out-of-coalition feature, where the line begins at a fixed baseline value and terminates on the feature’s true value. Each of these approaches involves evaluating the model on spliced data points that lie off the data manifold.
>
> To clarify this in our paper, we will provide a more thorough discussion of [1], including how our work is different – similar to that described above. We will also consider revising the sentence that the reviewer quoted in reference to this concern: “A performant method to estimate on-manifold Shapley values on general data is until-now lacking and a focus of this work.” While we stand by this statement, perhaps changing the word “estimate” to “compute” will make our contribution here clearer? (We are open to suggestions on this revision.)
>
> As a side note: We think it’s worth pointing out that, even though the off-manifold problem in explainability is indeed a known problem, our Section 3 presents many novel and precise findings on how this problem manifests itself that are until-now absent from the literature. We will enumerate these in our response to Reviewer 2.
>
> To respond to the reviewer’s two short notes:
> * We will change the title of our subsection from “A higher-dimensional example” to “Experiment on MNIST”.
> * The reviewer points out that available generative modelling techniques might struggle to learn the requisite conditional distributions on truly high dimensional data like ImageNet. Our supervised approach (Sec 4.1) – which bypasses the need for generating images and calculates the value function directly – was designed to address this exact problem highlighted by the reviewer.
>
> Having addressed the reviewer’s primary concern, we hope the reviewer will consider an increased scoring of our work.
>
> [1] M. Sundararajan & A. Najmi, “The Many Shapley Values for Model Explanation”, ICML 2020.

---

### Author Response · Authors · 2020-11-17
**Revision uploaded**

We would like to thank each reviewer for their valuable feedback. We have just uploaded a revision of our paper. In the revision, we have:

* clarified several points of confusion;
* included discussions of related works that we had initially missed;
* moved Figure 6 and Table 1 into the main body of our paper, as these provide important comparisons of our supervised and unsupervised methods;

all as detailed in our responses to individual reviewers. We believe this review process has improved our paper and hope that our readers agree.

---

### Author Response · Authors · 2020-11-20
**Columns added to Table 1 ("Off manifold" & "Empirical") to allow for comparison with baseline & optimal performance**

Following the excellent suggestion of Reviewer 4, we have added an "Off manifold" column to Table 1. This gives a sense of how significant the difference is between on- and off-manifold performance. We've also added an "Empirical" column to clarify (in the one case where the optimal MSE is empirically accessible) that an MSE of zero is not theoretically achievable in this context: there is irreducible error in predicting a model's output given only its partial input.

We've also added corresponding discussion to the first few paragraphs of page 9.

We hope reviewers agree that this addition further improves our paper.

---

### Decision · Program_Chairs · 2021-01-07
**Final Decision**

**Decision:**

Accept (Poster)

**Comment:**

A good paper with significant contribution on XAI and the on- vs off- data manifold explainability.
Reviewers have appreciated authors’ feedback and update of the paper (R1, R2, R4). I would like to personally thank the authors for a smooth, extensive and focused interaction w/ updates.